# How Do Runners Experience Personalization of Their Training Scheme: The Inspirun E-Coach?

**DOI:** 10.3390/s20164590

**Published:** 2020-08-15

**Authors:** Mark Janssen, Jos Goudsmit, Coen Lauwerijssen, Aarnout Brombacher, Carine Lallemand, Steven Vos

**Affiliations:** 1School of Sport Studies, Fontys University of Applied Science, 5644 HZ Eindhoven, The Netherlands; j.goudsmit@fontys.nl (J.G.); s.vos@tue.nl (S.V.); 2Department of Industrial Design, Eindhoven University of Technology, 5612 AZ Eindhoven, The Netherlands; c.e.lallemand@tue.nl (C.L.); a.c.brombacher@tue.nl (A.B.); 32M Engineering Ltd., 5555 XC Valkenswaard, The Netherlands; lauwerijssen@2mel.nl; 4HCI Research Group, Department of Behavioural and Cognitive Sciences, University of Luxembourg, 4366 Luxembourg, Luxembourg

**Keywords:** personalization, app, e-coaching, tailoring, running, training, workload, experience, heart rate, RPE

## Abstract

Among runners, there is a high drop-out rate due to injuries and loss of motivation. These runners often lack personalized guidance and support. While there is much potential for sports apps to act as (e-)coaches to help these runners to avoid injuries, set goals, and maintain good intentions, most available running apps primarily focus on persuasive design features like monitoring, they offer few or no features that support personalized guidance (e.g., personalized training schemes). Therefore, we give a detailed description of the working mechanism of Inspirun e-Coach app and on how this app uses a personalized coaching approach with automatic adaptation of training schemes based on biofeedback and GPS-data. We also share insights into how end-users experience this working mechanism. The primary conclusion of this study is that the working mechanism (if provided with accurate data) automatically adapts training sessions to the runners’ physical workload and stimulates runners’ goal perception, motivation, and experienced personalization. With this mechanism, we attempted to make optimal use of the potential of wearable technology to support the large group of novice or less experienced runners and that by providing insight in our working mechanisms, it can be applied in other technologies, wearables, and types of sports.

## 1. Introduction

In recent years there has been an exponential increase in the availability and use of sports and physical activity-related monitoring devices such as smartphone applications (apps), activity trackers, and sports watches [1,2]. This increased use of these monitoring devices is consistent with trends like Quantified Self [3] and mHealth [4], which emphasize the potential of these monitoring devices to contribute to a healthy and active lifestyle by supporting behavior change [5]. In particular, smartphones have several advantages. They are widely used, are embedded in everyday life [6,7], and allow people to collect data anywhere, anytime [8]. Since most people already have a smartphone (up to 76% of adults) [9] and apps are relatively cheap, often even free of charge, apps are accessible for almost everyone. The use of sports apps in Western Europe is mainly reflected in individual, recreational sports such as running, cycling, walking, and fitness. Mainly among runners, apps are widely used. Research shows that approximately 50–75% of (event) runners use a running app, especially novice or less experienced runners [1]. Among these runners, there is a high drop-out rate due to injuries and loss of motivation. Because these runners often lack personalized guidance and support. While there is much potential for sports apps to act as (e-)coach to help these runners to avoid injuries, set goals, and maintain good intentions [6,10,11]. While most available running apps primarily focus on persuasive design features like monitoring, they offer few or no features that support personalized guidance (e.g., personalized training schemes) [2,12].

Vos et al. [6] designed Inspirun, an e-coach for runners. Inspirun is a running app that offers a personalized coaching approach with the automatic adaptation of training schemes based on biofeedback and GPS-data. In the study of Vos et al. [6] a full description is given on how they designed this app, and the steps they took to develop the app. Whereas, the present paper gives a detailed description of the working mechanism; the personalized coaching approach with the automatic adaptation of training schemes based on biofeedback and GPS-data. But also aims to give insight into how end-users experience this working mechanism.

The paper is organized as followed. First, we provide a short overview of related work. Second, we shortly describe the development and working mechanism of Inspirun. Third, we introduce the study protocol used to test Inspirun. Fourth, we present the results of the end-user test. Finally, we discuss the results of the study and present suggestions for future work.

## 2. Related Work

Persuasive technology is studied extensively in the literature, also in relation to apps. A framework called Persuasive Systems Design (PSD) model is widely used for designing and evaluating systems that influence the attitudes or behaviors of users [13]. A review of Matthews et al. [12] on persuasive technologies used in apps, concludes that the most commonly used persuasive feature was self-monitoring. Thereby, apps often use data that is collected by the app to motivate the user to stay engaged (i.e., rewards, reminders, and suggestions). Matthews and colleagues [12] also conclude that many proven persuasive features are not utilized. Personalization is an example of one of these features that is (often) not implemented. Personalization is described by PSD framework as offering personalized content and services to the user. So, a tailored or personalized feature is one that is adapted to the characteristics of the end-user [14]. Specifically for running, Van Hooren et al. [11] propose a framework to optimize real-time feedback for reducing injury risk and improving performance and motivation. They argue that personalized real-time feedback on workload and running technique can be provided based on the individual preferences, experiences, and motives. For example, personalizing the type of feedback to suit preferences of the runner or personalizing the runners’ training session to suit the runners’ workload capacity. Research on the development of running (and other sports) related apps include systems that use several persuasive features as described in the PSD model. These technological systems can be divided into three different groups of studies, based on their objectives. 

The first group of studies focuses on improving running technique to minimize injury. For example, Aranki et al. [15] developed RunningCoach, a mobile health system that monitors and gives feedback on running cadence to optimize it, and (possibly) minimize running injury. They use self-monitoring and a way of tailoring (feedback can be changed by the user) as a persuasive feature in their system. Another example is Runmerge, an app developed by Kiss et al. [16], which enhances body awareness using visualization of their steps to help runners towards a better running experience. Authors found that enhanced proprioception (i.e., ‘knowing your body’) can be beneficial for everyday running training. Nylander and Tholander [17] developed Runright, which provides real-time visual and audio feedback about the current running rhythm. Their non-interpretive visualization led users to their interpretation of the feedback. Valsted et al. [18] developed Strive, a wearable that aims to assist runners in achieving rhythmic breathing; a breathing technique that potentially leads to improved running results and lower injury risk. The user’s understanding of feedback patterns based on self-monitoring was assessed.

A second topic that is often studied is the social aspects of running. In this category, dialogue and social support features are common. For example, Timmerman [19], investigated how technology can support a group of runners. In line with that, Mueller et al. [20] investigated how technology can support the social aspects of running. By introducing ‘Jogging over a Distance’, a system that allowed runners all over the world to run together using an audio-based social comparison feature. Further, HeartLink [21], a system that broadcasts live biometric data to social networks, and RUFUS [22], a system that enabled runners to communicate with supporters using ‘praise’ during races, are examples that also focus on the social aspects of running.

The third group of studies aims to enhance the motivation of runners. For example, the e-coaching ecosystem [23] offers interactions between end-users and human trainers to enhance motivation and stimulate a healthy and active lifestyle. Again, social support features are used as part of persuasive technology. A human trainer was essential in this design. Runners were more engaged when professionals offered or supervised the training sessions, compared to a group of runners with self-made training sessions and no supervision. Whereas most work related to motivation is focused on novice runners, Knaving et al. [24] determined a framework and guidelines to design technology for experienced runners. 

In conclusion, most of the discussed studies do use persuasive features, mostly social support or dialogue support features (as classified in the PSD-model). None of them implement personalization features in general, and none of them aim to personalize training sessions based on the runners’ workload capacity. 

## 3. Development and Working Mechanisms of Inspirun

### 3.1. Development from a Multi-Disciplinary Perspective

Inspirun is an Android-based system built with Ionic Framework using JavaScript and AngularJS. The app is connected (reverse engineering of the API) to the heart rate monitor (Wahoo TICKR X, Atlanta, GA, USA) to collect the heart rate during running.

Inspirun was designed by a multidisciplinary team. This team was composed of experts in behavioral sciences (*n* = 1), human movement sciences (*n* = 2), electronic engineering (*n* = 2), and industrial design (*n* = 1). All experts were selected based on their educational background and having at least 5 years of experience in this domain. To understand the runner and to serve their interest and needs, crossovers between these different fields of work were necessary. The multidisciplinary design approach and the steps taken are reported in previously published work [6]. In this paper, we extend and elaborate on this work, through an in-depth analysis of the personalized coaching approach, including the automatic adaptation of training schemes based on biofeedback and GPS data. 

### 3.2. Algorithm That Automatically Adjusts the Training Schemes

To create an algorithm that automatically adjusts the training scheme, we analyzed the approach that experienced coaches and trainers use when creating training schedules for runners. Interviews with experienced coaches and trainers (those who had at least 5 years of experience in coaching and training) revealed that in general, most coaches and trainers take the following steps (see also Figure 1).

Collect data about the runners’ current running levelSelect running goals for the upcoming training period and analyze the data of the current running level.Select training sessions that match their level and contribute to the running goals.Monitor during the running session and coach during/after the running session (comparing the executed data to the prescribed data).Adjust the next training session based on the comparison of prescribed and executed data.Continue iteratively with step 4 and step 5.

#### 3.2.1. Steps 1 and 2: Data and Analysis of the Current Running Level and Select Running Goals

To collect the runners’ current running level and select running goals for the upcoming training period (Step 1), we first designed a questionnaire to give insight into their current running level and running goals. Runners had to choose one of four possible answers: ‘I am completely new to running’, ‘I can run for 15 min without walking’, ‘I can run for 30 min without walking’, or ‘I can run for 60 min without walking’. Next, the running goal selection in our system was based on this self-declared current running level. If they were completely new to running, the easiest running goal (running 15 min without walking) was automatically assigned. If they were already able to run for 15 min without walking, again a running goal was automatically assigned, namely running 5 km without walking. If a runner was able to run 30 or 60 min without walking, he/she was allowed to choose their own goal. Available options were (i) running 5 km faster, (ii) running 10 km without walking, or (iii) running 10 km faster (see Figure 2 for a simplified flowchart and Appendix A for the full detailed flowchart). 

In order to collect objective data about runners’ current running level, we designed different test programs that consist of three running sessions. For each test session, we collected the heart rate (body feedback), GPS-data, and the perception of the training intensity (Rating of Perceived Exertion (RPE-score), a subjective parameter). In the test sessions, Inspirun only gave instructions to the runner but no feedback. For example, in the first session one of the instructions was: ‘start running at a comfortable pace’ and ‘you are doing well if you breathe heavily but are able to have a conversation as well’. With this instruction, runners started running at their own comfortable pace. Meanwhile, the app registered heart rate data, running speed, and RPE-score. During the test sessions, information was collected on the heart rate values and RPE scores for different running speeds (jogging, easy running, comfortable speed, hard running, and very hard running) We labeled this relation between speed, heart rate, and RPE as the Personal Running Profile (PRP). See Table 1 for an example, of a current PRP after the three test sessions.

#### 3.2.2. Step 3: Select Training Sessions that Match their Level and Contribute to the Running Goals

The second step in the process was the selection of training sessions that match the runner’s current running level and that contribute to their goal. We used five generally accepted training principles (i.e., individualization, progression, overload, variation, and objective and subjective monitoring of the performance [25,26,27]) in combination with the expertise of the human movement scientists, involved in the development of the Inspirun e-coach, to create training schedules. For each goal (e.g., running 5 km without walking), a training schedule was constructed. In total, a schedule consists of 20 sessions divided over several weeks dependent on the number of sessions per week (between 1 and 3). To make the training schedules and sessions applicable to all runners, we chose to personalize training sessions based on workload and intensity. This means that the training schedule (i.e., distance, total time, and type of training) was the same for a runner with the same running level and goal, while the intensity varies per runner. We chose RPE as the parameter for intensity because of its validity, reliability, and internal consistency [28,29]. In the app, there is an explanation how runners should use the RPE score. For each score the runner can read an explanation how this RPE should feel in terms of breathing, the ability to talk, and the RPE is expressed in words like, hard/very hard/comfortable [29]. See Table 2 for some examples of training sessions.

#### 3.2.3. Step 4: Monitoring during the Running Session and Coaching during/after the Running Session (Comparing the Executed data to the Prescribed Data)

The third step is to monitor each running session and coach the runner (by comparing the performance data to the prescribed data). To prescribe a training session, the most current PRP (current PRP is an average of the last six sessions) is used for each session (see Table 3 for an example). This means that for each RPE present in that specific session, the matching speed and heart rate are selected.

For example, in Table 4, all data of session 7 of participant 7 is shown. Session 7 consists of blocks between RPE 3 and 8. The current PRP (see Table 3, PRP before session 7) of this participant is used to prescribe the speed and heart rate at a given RPE (e.g., at RPE 4, HR 124 bpm, speed 10.1 km/h). 

To coach the runner during the run, prescribed data and real-life data are constantly compared. This means that, for instance, when accordingly, the PRP of the prescribed heart rate in the RPE7 block should be 141 bpm, but real-life data shows heart rates that deviate more than 5 bpm from the prescribed heart rate, the app reacts with feedback. This feedback is given in a natural way, by giving the instructions to increase or decrease running speed. If a runner, in this example, has a real-life heart rate of 154 bpm, he/she is instructed to slow down, to lower the heart rate to the prescribed range of 136–146 bpm (141 ± 5 bpm). Whereas if the runner runs with for example 133 bpm, he/she is instructed to speed up, in order to increase the heart rate to the prescribed range of 136–146bpm (141 ± 5 bpm). Immediately after the training session, Inspirun asks the runner to fill in the RPE. In this case, it is expected that the runner fills in RPE of 7 for this block. If he/she does this (i.e., the filled-in RPE matches with the prescribed RPE), then the data from that running block will be used to improve the data of RPE7 in the PRP. If not, and for instance, RPE of 8 is given (while RPE7 was prescribed), the data of that running block is used in the data of RPE8 in the PRP.

After each training session, the system calculates a compliance-score based on the match of the actual monitored data (i.e., the heart rate and the mean running speed of the training blocks) with the prescribed data. A score of 100% means that the monitored data matched completely with the prescribed data. A score above 100% indicates that a runner performed better than expected (e.g., lower heart rate at a given speed or lower perceived intensity/higher running speed at prescribed heart rate), while a score below 100% means that a runner did not perform as well as expected. When calculating the compliance-score, only training blocks with an RPE of 4 or higher are taken into account, because an RPE score of 3 or lower is only used as recovery between intervals. Therefore, the first minute of every interval is not used in the calculation. The reason for this is that increase and decrease in heart rate was delayed compared to speed. After 60 s, heart rate should be leveled off and/or in steady-state (depending on the intensity) and then the relation to speed is meaningful and useful to calculate the match between prescribed and actual performance. Eventually, this compliance score gives insight into the progression of the runner and feedback for the runner how well the training session was executed. See Table 4 for the compliance scores of participant 7. This runner scored a total compliance-score of 97%, which indicates that the actual heart rate and speed were in line with the current PRP. 

#### 3.2.4. Step 5: Adjust the Next Training Session Based on the Comparison of Prescribed and Executed Data

In the final step, the data of the newly completed training is added to the PRP. The average heart rate and the average speed per training block are computed. In line with the calculation of the compliance-score, the first minute is not taken into account. 

We illustrate this with the example from Table 4. Towards the end of the session, there is a training block of 4 min on RPE7 (which is highlighted). The average speed is calculated by averaging 9.54, 9.35, and 8.41 km/h, and the heart rate by averaging 141, 144, 145. In this case, resulting in 9.1 km/h and 143 bpm (assuming the runner did perceive this running block as RPE of 7). This 9.1 km/h and 143 bpm at RPE 7 are added to the PRP. 

Before the start of every new session, the data of the last six training sessions are used to calculate average heart rates and average speeds per RPE (as seen in Table 3). These values are labeled as the current PRP. In the case of participant 7 (illustrated in Table 3), the PRP after session 7, uses the data of session 2, 3, 4, 5, 6, and 7, (which are the last 6 sessions) and the data of session 1 is not included in the PRP anymore. By doing this, the PRP is constantly updated and the runners’ profile is continuously evaluated. Furthermore, as a consequence of using an average over six sessions, outliers (i.e., an exceptional good or bad training session) have minimal impact on the PRP. However, as a runner improves over time, the PRP gradually adjusts, and thus, the prescribed intensity for the next training session changes gradually, minimizing the risk of injury [30]. 

## 4. Study Protocol

### 4.1. Study Protocol

We designed a study protocol to get insight into (i) how end-users experience the personalization of the training schedule, and (ii) whether this approach motivates them to keep running. Between spring 2018 and autumn 2018, we posted a call on various social media to participate in our study. All runners that were injury-free and were willing to train for one of the five goals Inspirun focusses on could participate. The participants had to (i) run at least once a week and (ii) use the Inspirun app until they completed a training schedule consisting of 20 training sessions. We used online questionnaires and the data collected by the Inspirun app to monitor the participants over time. Ethical approval for the study was obtained by the Ethical Research Committee of Fontys University of Applied Sciences. An introduction letter informed the participants about the purpose of the study, the anonymization of the data, and the incentive (Wahoo TICKR X) they would receive when completing the 20 training sessions. In total, 43 runners reacted that they want to participate in our study, of which 19 participants agreed to participate in the study and complete all training sessions. 

Before their first run, participants had to complete the first questionnaire (T0) to gather information on their socio-demographics, running experience, and previous experience with apps and wearable technology. After three running (test) sessions (T1), participants received a second questionnaire every five sessions (T1, T2, T3, T4, T5). This questionnaire focused on their experiences with the app over the last sessions. Participants continued to receive this questionnaire until they had completed the training schedule (=20 sessions). After completion of the training schedule (T5), an additional third questionnaire (T6) was provided to score their experience over the full test period (see Table 5). Overall, we thus covered the different timespans of user experience; anticipated, episodic, and cumulative experience as defined by Roto et al. [31].

### 4.2. Measures

The first questionnaire (used at T0) was constructed similarly to previous research of Janssen et al. [1,32,33] and Clermont et al. [34] containing a set of variables including (i) socio-demographic variables; (ii) running-related variables; and (iii) previous experience with wearable technology. The socio-demographic variables include gender, age, and level of education. The group of running-related characteristics consists of variables that are directly related to running and that define the level of running involvement: running frequency (number of runs per week), running distance (distance in kilometer per week), active participation in running (years of active running participation), running context (individual, with friends, colleagues and/or running groups, or running clubs), and the most practiced sport (running/other sport). 

For T1–T5, a second questionnaire was used. With this questionnaire, we measured the user experience over the last five training sessions with three items (5-point Likert, ranging from 1 = completely disagree to 5 = completely agree) focused on goal perception, motivation to keep running, and personalization. In item 1, we asked whether they felt that the training sessions contributed to their goal. In item 2, whether Inspirun motivates them to run, and in item 3, whether the intensity of the training sessions was accurately adjusted to their running level. For all items, we gave them the option to explain why they gave that particular score (open-ended question). 

The third questionnaire (T6), retrospectively measured the entire experience of the automatically adjusted training sessions and motivation to keep running throughout all training sessions. We used the same three items as in the second questionnaire, but we asked the participants to assess these items over the entire testing period instead of the last five training sessions.

### 4.3. Analysis

Frequencies and descriptive analyses were run on (i) socio-demographic variables; (ii) running-related variables; and (iii) experience with technology. Secondly, descriptive analyses (mean, sd) for all three items, for every questionnaire (T1–T6) were calculated. Third, possible differences in these three items over time (between T1, T2, T3, T4, T5, and T6) were investigated using the non-parametric version of repeated measure (Friedman). Finally, Spearman correlations were analyzed between the three items using all answers (T1, T2, T3, T4, T5, and T6 combined).

## 5. Results

### 5.1. Participants

Ten of the 19 runners that completed the training schedule are female. The participants’ age ranged from 21 years old to 60 years old (averaging 35 years). Of which, 85% is higher educated, and 10 of them are fulltime employed. Among the 19 participants, the distribution over the different running goals was relatively equal (see Table 6). Although none of them belonged to the beginner group (those who could not run 15 min without walking), two of the 19 runners did not participate in running before (but were still capable of running 15 min without walking). Five of the 19 runners perceive running as their main sport. The running experience ranges from less than 3 months to 5 or more years of experience. Whereas most runners run 0 to 5 km per session and run once or twice a week. Finally, out of the 17 that ran before, 13 runners run mostly individually, the rest runs in a sports club, with friends, family, or small running groups.

### 5.2. Motivation and Goal Perception

Item 1 on goal perception and item 2 on motivation were both answered 111 times (T1–T6), the goal perception scored an average of 3.95 (SD = 0.88) and the motivation to keep running scored on average 4.01 (SD = 0.99) (see Table 7 and Figure 3).

On both items, no score of 1 out of 5 (totally disagree) was given. On item 1 goal perception, 9 times a negative score (disagree) was given, 17 times a neutral score of 3 out of 5, 55 times a positive score of 4 (agree), and 34 times the maximum positive score of 5 out of 5, meaning that they totally agreed with this item. On motivation (item 2), 11 times a negative score was given (again only scores of 2 out of 5), 20 times a neutral score of 3 out of 5, 36 times a positive score of 4, and 44 times the maximum positive score of 5 out of 5. The optional open-ended questions revealed that overall, most runners perceived the sessions as challenging but not too hard. They also had the feeling that they (slowly) progressed over time. Some runners perceived this as positive, while others as negative. In contrast, the negative influencers of the motivation were mostly the irritation of trying to reconnect the heart rate sensor while running and the summary after the run when it gave incorrect speed or heart rate due to measuring flaws. 

### 5.3. Personalization 

On average (*N* = 111), the personalization of the session with regards to the running level was assessed as 3.84 (SD 0.82) on a 5-point Likert scale (Table 7).

Out of 111 times this item was answered, 11 times a negative score was given and 13 times a neutral score of 3 out of 5. Most participants explained these scores. Reasons mentioned were related to the accuracy of the speed measurement, that the speed was not measured accuracy and that therefore they had the feeling that the scheme could not personalize accordingly. Therefore, also instabilities of the Bluetooth connection between the phone and the heart rate monitor was given as a reason, therefore heart rate connection was sometimes lost during running, causing inaccurate heart rate measures. We checked these claims by analyzing the app data and indeed, for these particular runners, the heart rate or GPS data was missing from the PRP for multiple runs. Finally, the runners mentioned that some sessions were too easy and that they had the feeling that the app was adjusting too slow or being too conservative. They felt they could progress faster than the algorithm prescribed. Another explanation why they scored negatively or neutral mentioned that they expected a different personalization, for example on the type of session (i.e., interval or endurance training) and on time and distance (i.e., self-selecting the distance, instead of prescribed by the app), rather than adjustments on intensity. Those who were positive (*n* = 87) about the personalization (scoring a 4 or 5) experienced the gradual increase of the intensity as pleasant. They (again) stated that the sessions were challenging without being too hard (Figure 2). 

### 5.4. Relation between Personalization, Motivation, and Goal Perception

For all three items, there was no significant difference between anticipated, episodic, and cumulative experiences, meaning that no differences were found across time during the testing period. Spearman-correlations (*n* = 111) revealed a moderate to strong positive relation between items. Goal perception is moderately correlated to the motivation to keep running, with an explained variance of 48% (*r* = 0.695, *p* < 0.001). Goal perception also correlates moderately with personalization, with an explained variance of 36% (*r* = 0.597, *p* < 0.001). While motivation to keep running correlates also moderately with personalization, with an explained variance of 32% (*r* = 0.563, *p* < 0.001). These analyses show that if scores on one of the items increase, in 32% to 48% of the cases, the score on the other items increases too. 

## 6. Discussion and Conclusions 

### 6.1. Personalization of Training Session

Inspirun was designed to provide personalized training schemes based on biofeedback, GPS-data, and RPE. The present paper gave a detailed description of the working mechanism; the personalized coaching approach with the automatic adaptation of training schemes based on biofeedback and GPS-data, and aimed to give insight into how end-users experience this working.

The primary conclusion of this study is that the working mechanism (if provided with accurate data) automatically adapts training sessions to the runners’ physical workload and stimulates runners’ goal perception, motivation, and experienced personalization. 

Our user study revealed that in general, the personalization of the intensity of the training sessions was experienced as personalized by the participants. Participants, whose data collected with the app was complete and without flaws and gaps, found that the sessions were accurate, and experienced them as personalized. We expected that using generally accepted training principles (i.e., individualization, progression, overload, variation, and objective and subjective monitoring) to personalize and adjust the session would works. In line with Van Hooren et al. [11] who stated that workload can be used as characteristics to personalize. Indeed, our findings confirm that building a feature that personalizes the workload (PRP) on both subjective (RPE) and objective (HR and Speed) aspects of intensity seems to be a good combination to develop a mechanism that is sensible for change and is robust enough to deal with fluctuation between sessions. We used personalization as a persuasive feature derived from the Persuasive Systems Design (PSD) model [13], in which personalization is a feature that is not often used in persuasive design, despite its potential [12].

For runners whose data collected by the app was incomplete or inaccurate, the mechanism was not perceived as personalized. This is in line with challenges described in previous research, for example, Seshadri et al. [35] showed that when developing sensor technology we should design robust and easy-to-wear systems with improved signal to noise ratios. Wan an et al. [36] stated that there are still many challenges before the actual implementation of technology in practical applications, they also refer to noisy signals and challenging issues regarding the power source. Specht et al. [37] investigated satellite-based solutions to identify positioning (time, distance, and speed) and found that the existing systems (e.g., GPS) are not accurate. Especially when more intense activities are performed, the processing of raw data needs to improve in order to provide accurate and meaningful feedback [38]. In our case, running can be classified as intense activity-gaps in heart rate data caused problems in the automatic calculation of the next session. 

For example, some runners had problems with the Bluetooth connection between the heart rate monitor and their phone. While running, the Bluetooth connection was lost, and the app did not collect heart rate data. Instead it writes zeros in the dataset. In most cases, this caused a much lower average heart rate over a training block (due to many zeros). Consequently, making the app think that the automatically generated heart rates were too high in relation to running speed and RPE. The same goes for inaccuracies in running speed. Runners experienced some problems when running in wooded and hilly areas, resulting in mostly lower speeds than expected, making the app think that the session was too challenging to complete when this was actually due to the environment (i.e., sandy surfaces, altimeters) accuracy of the GPS signal (i.e., cutting corners, densely vegetated areas). The mechanism of creating automatically and personalized training sessions thus seems to work when provided with complete and accurate data, although the robustness of the mechanism, especially how it deals with flaws in the dataset needs improvements. In future work, first of all, the Bluetooth connection must be improved. Second, to make the mechanism more robust, the algorithm must be adjusted so that incomplete datasets (e.g., with too many zeros) are not used or used differently in the PRP. 

### 6.2. Motivation to Keep Running

On average, the participants were quite positive about the influence on the app on their motivation to keep running. The open-ended question revealed that the reasons for scoring lower on motivation are essentially the same as the reasons that lowered the experienced personalization. In both cases, the bug that caused disconnection of the heart rate monitor and the resulting incomplete and inaccurate data were mentioned. Therefore, we expected that personalization and motivation influenced each other, both negatively as positively. Based on the Self Determination Theory [39] and the Fogg Behavior Model [40], we argue that sessions that match with the physical capacity of the runner are more fun to complete, giving the runner a sense of achievement, and increasing the motivation to keep running and challenging themselves. While sessions that are too easy or too hard are not fun to complete and negatively influence the intentions to keep running. 

A correlation revealed a relation between experience personalization (item 3) and motivation (item 2). If scores on experience personalization increased, in 32% of the cases the score on motivation increased too (or the other way around). As this only explained one-third of the variance, there must be more underlying reasons and variables that explain variances in the scores. We did not include items on injuries, yet we know from literature (e.g., [41]) that having an injury (i.e., being not fit) negatively influences the motivation. Finally, in future work, we see many opportunities to further improve the app by exploring features that could increase the autonomy of the runner or enhance social cohesion. Since for now the app strongly focusses on enhancing the (self) competence of the runner.

### 6.3. Limitations and Future Work

In this study, we faced some technical challenges, like the inaccuracy of the data, due to Bluetooth disconnections. A solution here might be technical improvements. We logged all data with the app, so hopefully, future analysis on that data can find the causes of the random disconnections. Another solution might be implementing a warning signal when data is inaccurate and that the runner must reconnect otherwise the app stops recording the training session. Or allowing the user to remove flawed data, using a more user-centered approach. Therefore, we could make the mechanism more robust, and the algorithm must be adjusted so that incomplete datasets (e.g., with too many zeros) are not used or used differently in the PRP. New ‘rules’ should be constructed and programmed before further testing among runners so time gaps between sessions should be taken into consideration. The Inspirun does not detect time gaps between sessions (e.g., four weeks of no training between session 5 and 6) and therefore it does not take into account potential detraining effects. Furthermore, in line with the third guideline ‘how to design for runners’ described by Knaving et al. [24], who advised building systems that strengthen the runners’ intrinsic motivations, which is the most autonomous form of motivation within the Self Determination Theory [39], we see a lot of opportunities to improve the personalization of the app by giving more autonomy to the runner or stimulate social-support. For example, the option to choose different types of training sessions, or choosing between two sessions that are preselected by the app or sharing sessions with others. Finally, in the current version, an average of six sessions is used to calculate the PRP, given the feedback of several runners that they thought that Inspirun was adjusting in a slightly too conservative (too slow) way, in future iterations the possibility to personalize how the Inspirun adjusts, especially for those runners who adapt better or faster to the stimuli could be explored. A solution here might be technical (improving the technology) or user-centered (giving control to a user to remove flawed data). 

Besides the above mentioned technical limitations and future work recommendations, there are also methodological recommendation for future work. In this study we aimed to give insight into how end-users experience the working mechanism of Inspirun. In future work it could be interesting to look into the effect of Inspirun on aspects such as performance, motivation, and injury-free running, compared to other interventions (such as other apps or standardized training schemes).

### 6.4. Conclusions

Inspirun was designed to provide personalized training schemes based on biofeedback, GPS-data, and RPE. The primary conclusion of this study is that the working mechanism (if provided with accurate data) automatically adapts training sessions to the runners’ physical workload and stimulates runners’ goal perception, motivation, and experienced personalization. 

With our work, we attempted to make optimal use of the potential of wearable technology to support runners. In particular, the large group of novice or less experienced runners who lack guidance. This work also contributes to the emerging area of designing for running and wearable technology trends by providing working mechanisms that are applicable to other technologies, wearables, and types of sports. Consequently, we hope that runners profit from technologies like the Inspirun to run injury-free and keep motivated.

## Figures and Tables

**Figure 1 sensors-20-04590-f001:**
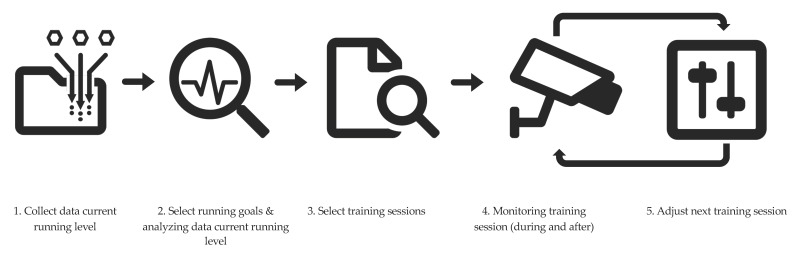
Steps taken by coaches and trainers when creating training schedules for runners. Final step is to continue iteratively with step 4 and step 5.

**Figure 2 sensors-20-04590-f002:**
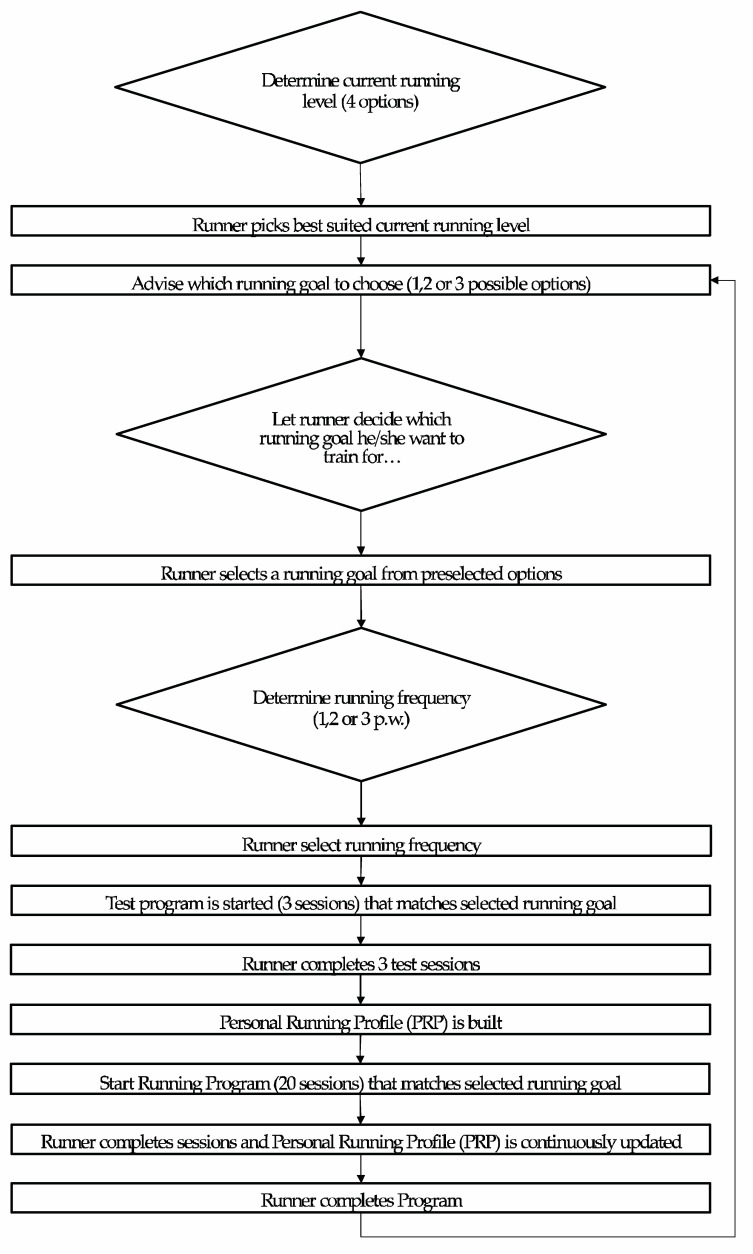
Simplified flowchart of Inspirun. Starting from the running experience, to running goal, and which training scheme fits best. For full details see Appendix A Detailed Flowchart.

**Figure 3 sensors-20-04590-f003:**
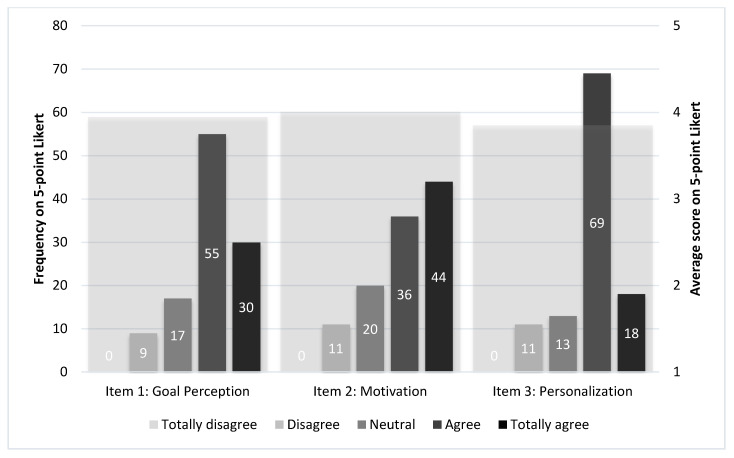
Frequency of given answer on a 5-point Likert scale on items 1, 2, and 3 (left *y*-as) and the average score of items 1, 2, and 3 (right *y*-as).

**Table 1 sensors-20-04590-t001:** Example of a current Personal Running Profile (PRP) after the three test sessions, heart rate values (beats per minute), and Rating of Perceived Exertion (RPE) scores for different running speeds (kilometers per hour) (jogging, easy running, comfortable speed, hard running, and very hard running).

	Test 1			Test 2			Test 3			Current PRP		
Pace	RPE	Speed	Heart rate	RPE	Speed	Heart rate	RPE	Speed	Heart rate	RPE	Speed	Heart rate
Jogging	4	10.0	121	4	10.2	124				4	10.1	123
Easy							5	11.6	139	5	11.6	139
Comfortable	6	12.3	145	6	12.2	146	6	12.1	143	6	12.2	145
Hard				7	13.8	156	7	13.4	150	7	13.6	153
Very Hard							8	14.8	173	8	14.8	173

**Table 2 sensors-20-04590-t002:** Examples of training sessions within a training scheme, different intensity blocks are highlighted. Illustrated sessions use different types of interval (short and long), have a duration between 27 and 32 min, and the intensity varies between RPE3 till RPE8.

Session	Time in Minutes per Session																															
	1	2	3	4	5	6	7	8	9	10	11	12	13	14	15	16	17	18	19	20	21	22	23	24	25	26	27	28	29	30	31	32
1	4	4	4	6	6	6	3	7	7	4	8	8	3	3	7	7	4	8	8	3	3	7	7	4	8	8	3	3	4	4	4	4
2	4	4	4	6	6	3	7	7	8	7	7	7	7	8	7	7	7	7	8	7	7	3	3	4	4	4	4					
3	4	4	4	6	6	6	3	6	6	6	6	6	6	6	6	7	3	3	6	6	6	6	6	6	6	6	7	3	4	4	4	4
4	4	4	4	6	6	3	6	6	6	6	6	7	7	7	6	6	6	6	7	7	7	6	6	6	3	4	4	4	4			
5	4	4	4	6	6	6	3	8	8	8	3	7	7	7	3	6	8	6	3	7	7	7	3	8	8	8	3	3	4	4	4	4
6	4	4	4	6	6	3	6	6	7	7	8	7	7	7	7	6	6	7	7	7	7	8	7	7	3	3	4	4	4	4		

**Table 3 sensors-20-04590-t003:** Current PRP before session 7 from participant 7. Heart rate in beats per minute, speed in kilometers per hour.

Current PRP before Session 7		
RPE	Speed	Heart Rate
1	0	0
2	7.20	115
3	7.74	120
4	7.85	124
5	7.99	125
6	9.00	133
7	10.01	141
8	11.02	155
9	11.99	169
10	12.24	173

**Table 4 sensors-20-04590-t004:** Overview of the data of session 7 from participant 7. Showing the prescribed intensity in RPE, speed (in kilometers per hour) and heart rate (in beats per minute) in the first columns, the actual monitored data (while running) in the middle columns, and the compliance scores in the last columns.

	Prescribed		Actual		Compliance		
Intensity	Speed	Heart Rate	Speed	Heart Rate	Speed	Heart Rate	Total
4	7.85	124	6.79	99	*	*	*
4	7.85	124	7.83	128	100%	103%	97%
4	7.85	124	8.10	132	103%	107%	97%
6	9.00	133	8.65	136	*	*	*
6	9.00	133	8.65	142	96%	107%	90%
3	7.74	120	6.09	125	**	**	**
5	7.99	125	8.28	121	*	*****	*****
5	7.99	125	9.73	146	122%	117%	104%
5	7.99	125	9.79	149	122%	119%	102%
5	7.99	125	9.85	148	123%	119%	104%
5	7.99	125	9.08	146	114%	117%	97%
5	7.99	125	9.30	148	116%	119%	98%
5	7.99	125	9.24	145	116%	116%	99%
3	7.74	120	5.63	125	**	**	**
3	7.74	120	5.17	99	**	**	**
6	9.00	133	8.54	130	*	*	*
6	9.00	133	9.83	145	109%	109%	100%
6	9.00	133	9.27	144	103%	108%	95%
6	9.00	133	10.01	148	111%	112%	100%
6	9.00	133	8.11	145	90%	109%	83%
6	9.00	133	9.43	143	105%	107%	98%
6	9.00	133	9.53	147	106%	111%	96%
3	7.74	120	5.75	129	**	**	**
3	7.74	120	5.13	100	**	**	**
7	10.01	141	7.52	113	*	*	*
7	10.01	141	9.54	141	95%	100%	95%
7	10.01	141	9.35	144	93%	102%	91%
7	10.01	141	8.41	145	84%	103%	82%
3	7.74	120	6.33	129	**	**	**
3	7.74	120	6.02	111	**	**	**
8	11.02	155	12.09	145	*	*	*
8	11.02	155	13.49	169	122%	109%	112%
8	11.02	155	12.78	171	116%	110%	105%
8	11.02	155	12.26	170	111%	110%	102%
3	7.74	120	6.46	154	**	**	**
4	7.85	124	7.19	132	*	*	*
4	7.85	124	7.75	137	99%	110%	89%
4	7.85	124	8.18	138	104%	111%	94%
4	7.85	124	8.15	141	104%	114%	91%
4	7.85	124	7.96	139	101%	112%	90%
5.66	8.87	132.75	9.21	141.51	107%	110%	97%

* Not calculated since it is the first minute this block. ** Not calculated since RPE < 4.

**Table 5 sensors-20-04590-t005:** Timeline of the three questionnaires (Q1, Q2, and Q3).

	T0	T1	T2	T3	T4	T5	T6
Timespan of experience	Anticipated experience	Episodic experience					Cumulative experience
When	Before using the apps	After 3 test sessions	After 5 training sessions	After 10 training sessions	After 15 training sessions	After 20 training sessions	After 20 training sessions
Question-naire	Q1: Background information	Q2: Experience with the app over the last sessions					Q3: Experience with the app over the full test period
Number of responses	19	19	19	19	19	19	16

**Table 6 sensors-20-04590-t006:** Overview of all descriptive variables for each participant of the first questionnaire.

Participant	Goal	Age	Gender	Education	Currently a Runner	Main Sport	Active Participation in Running	Average Training Distance per Run	Frequency per Week	Running Context
1	05 km	42	Male	Middle	Yes	No	<1 year	0–5 km	2×	Individually
2	05 km	21	Female	Higher	Yes	No	<3 months	0–5 km	1× a year	Individually
3	05 km	48	Male	Higher	Yes	No	<3 months	0–5 km	1× a year	Individually
4	05 km	40	Female	Middle	No					
5	05 km	26	Female	Higher	No					
6	05 kmfaster	26	Female	Higher	Yes	No	1–3 years	0–5 km	1×	Individually
7	05 kmfaster	41	Male	Higher	Yes	Yes	1–3 years	6–10 km	1×	Small Running Groups
8	05 kmfaster	33	Female	Higher	Yes	No	1–3 years	0–5 km	1×	Individually
9	05 kmfaster	37	Female	Higher	Yes	Yes	1–3 years	6–10 km	2×	Friends and Family
10	05 kmfaster	23	Male	Middle	Yes	No	<3 months	0–5 km	1×	Individually
11	10 km	22	Male	Higher	Yes	No	1–3 years	6–10 km	1×	Individually
12	10 km	60	Male	Higher	Yes	No	1–3 years	0–5 km	2×	Friends and Family
13	10 km	21	Female	Higher	Yes	No	<3 months	0–5 km	1×	Individually
14	10 km	43	Male	Higher	Yes	No	>5 years	0–5 km	1× a month	Individually
15	10 km	41	Female	Higher	Yes	Yes	>5 years	0–5 km	2×	Small Running Groups
16	10 kmfaster	30	Female	Higher	Yes	Yes	1–3 years	6–10 km	3×	Individually
17	10 kmfaster	35	Male	Higher	Yes	No	1–3 years	6–10 km	1×	Individually
18	10 kmfaster	42	Female	Higher	Yes	No	>5 years	11–15 km	2×	Individually
19	10 kmfaster	43	Male	Higher	Yes	Yes	>5 years	11–15 km	1×	Individually

**Table 7 sensors-20-04590-t007:** Mean scores and standard deviations on all items.

	Item 1: Goal Perceptions		Item 2: Motivation		Item 3: Personalization	
	Mean	SD	Mean	SD	Mean	SD
T1	4.05	0.85	4.37	0.83	3.95	0.91
T2	3.95	0.78	3.95	1.03	3.95	0.52
T3	4.16	0.83	4.11	1.10	3.84	0.90
T4	3.95	0.85	4.00	0.94	3.79	0.79
T5	3.84	0.83	3.84	0.96	3.84	0.90
T6	3.75	1.13	3.81	1.11	3.69	0.87
Average	3.95	0.88	4.01	0.99	3.84	0.82

All tested for Friedman repeated measures, no sign between T1–T6 for any item.

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
