# Peer review of "How Do Runners Experience Personalization of Their Training Scheme: The Inspirun E-Coach?"

_sensors, 2020, doi:10.3390/s20164590_

Round 1

Reviewer 1 Report

Review Sensors-889337– “How do Runners Experience Personalization of Training Scheme: the Inspirun E-Coach?

This manuscript explains how the Inspirun app works and conducts a small study to investigate if the apps helps motivate runners. 

The study protocol has many flaws.  There is no control group to compare the app users to.  There are also no statistics used in this study to attempt to answer the authors questions about whether or not the app helps motivate the runners.  Even with no control group the authors could compare motivation before and after implementing the app into their running program.

Specific Comments

 Line 181 - Figure 2.   Needs to be changed to two figures (or another solution) to make the flow chart easier for readers to read.

Line 260 - Study Protocol - Who do you recruit for the study?  What were your inclusion/exclusion criteria?

Line 303 – Analysis.  These are not the appropriate analysis for this study design.

Reviewer 2 Report

Manuscript title

How do Runners Experience Personalization of their 3 Training Scheme: the Inspirun E-Coach?

The paper aim was to detail the running app, Inspirun E-coach app by describing its working mechanism; the personalized coaching approach with the automatic adaptation of training schemes based on biofeedback and GPS-data from the app.

Major concerns

- Line 170. What are the “generally accepted training principles”.

- Figure 1 which is an important part of the paper is too small to be read. This reviewer is not able to decipher what was written in Figure 1 – reviewer is not able to read nothing at all. Please re-do.

- Line178. “we chose RPE as the parameter for intensity”. While this point is valid, this based on the assumption that the runner knows how to provide a valid RPE value to his exercise intensity – is there an explanation in the app on teaching an individual how to ‘assess’ his exercise intensity using the RPE? There is a need to quantify the ‘changes’ for the training scheme after the 6 training sessions.

- For the testing and training variables/ indictors (Table 1) Please provide justification for using relative  heart rate and relative speeds. This reviewer feel it is more appropriate to use % of the individual’s maximal heart rate rather than absolute heart rate values as well as relative speed (as a percentage of the individual’s maximal aerobic speed) rather than absolute speed.

Minor concerns

- Line 120-122. On what basis/criteria does the authors proclaim that the individuals are “experts”? Some criterion measure of experts needs to be written and define.

- Line 129-130. Again, on what basis/criteria does the authors proclaim that these individuals are “experts”? Some criterion measure of experts needs to be written and define.

- Line 251. Why the last 6 training sessions? What happen is the gap or time period between the training sessions is “extended”, for example, the period between training session no 5 and no. 6 was > 4 weeks in duration; would the app able to detect this and eliminate the data taken from the training session no. 6?

- Line 385. I don’t think with the results that you have achieved that you can claim that the app is “accurate”.

- Line 356. Because the current app seems to provide training programmes that is too conservative or too easy, suggest that the app’s automatic adaptation provides 3 different training schemes after every 6 training sessions to choose from. One training scheme that is conservative in its intensity and duration, another that is neutral and another is on the heavy side or aggressive training scheme in terms of intensity and duration – and let the individual choose which one he/she wants to do. And over time, the app will be able to learn whether to provide a conservative or aggressive training scheme for this same individual.

- maybe your app should have a system within it like a warning signal or noise to the individual when HR data is missing (disconnection) after a continuous period 60 s – so that action can be taken to rectify the problem on the spot so that there is not a lot of missing data during the exercise.

Round 2

Reviewer 1 Report

The authors provided a thorough rational for their study design, improved the manuscript, and addressed all of my concerns.

Reviewer 2 Report

Reviewer is satisfied with the changes made.